# Association Between Advanced TNM Stages and Increased Risk of Cardiac Dysfunction in Patients with LVEF < 50%

**DOI:** 10.3390/medicina61020301

**Published:** 2025-02-10

**Authors:** Sergiu Ioan Murg, Loredana Matiș, Andrada Florina Moldovan, Dorel Ionel Popovici, Alina Gabriela Negru, Timea Claudia Ghitea, Mircea Ioachim Popescu

**Affiliations:** 1Doctoral School, Faculty of Medicine and Pharmacy, University of Oradea, 410068 Oradea, Romania; sergiumurg@yahoo.com; 2Department of Clinical Discipline, Faculty of Medicine and Pharmacy, University of Oradea, 410068 Oradea, Romania; matisloredana@yahoo.com (L.M.); onita.andrada@yahoo.com (A.F.M.); procardia_oradea@yahoo.com (M.I.P.); 3Department of Oncology, Faculty of Medicine, Victor Babeş University of Medicine and Pharmacy Timisoara, Eftimie Murgu Square 2, 300041 Timisoara, Romania; dorel.popovici@umft.ro; 4Department of Cardiovascular Diseases of Timisoara, Victor Babeş University of Medicine and Pharmacy Timisoara, G. Adam Str. No. 13A, 300310 Timisoara, Romania; alinanegru@umft.ro; 5Pharmacy Department, Faculty of Medicine and Pharmacy, University of Oradea, 410068 Oradea, Romania

**Keywords:** HER2-targeted therapies, cardiotoxicity, cancer therapies, left ventricular ejection fraction (LVEF), chemotherapy-induced heart failure

## Abstract

*Background and Objectives:* Cardio-oncology addresses the growing concern of cardiovascular complications arising from cancer therapies. Although cancer treatments have greatly enhanced survival outcomes, they frequently carry substantial risks to cardiovascular health. This research examines the cardiovascular toxicity associated with HER2-targeted therapies, focusing on the interconnection between tumor characteristics, including histopathological profiles and TNM classification, and the development of cardiovascular complications. The objective is to identify key correlations that inform better prevention and management strategies for cardiotoxicity in oncology patients. *Materials and Methods:* This retrospective study analyzed cancer patients undergoing cytostatic treatments, particularly anthracyclines, radiotherapy, and HER2-targeted therapies. Cardiac function was monitored using echocardiographic assessments, including global longitudinal strain and left ventricular ejection fraction (LVEF). Patients were stratified based on TNM cancer staging and histopathological findings to evaluate correlations between treatment regimens and cardiovascular outcomes. *Results:* The analysis revealed a significant association between advanced TNM stages and reduced LVEF, with patients in stage T4 showing the highest prevalence of cardiac dysfunction. Cytostatic treatments, such as anthracyclines and HER2-targeted therapies, were identified as key contributors to cardiotoxicity, particularly in advanced-stage cancer patients. These findings emphasize the importance of regular cardiac monitoring to detect early signs of cardiotoxicity, as patients with pre-existing cardiovascular risk factors demonstrated a higher prevalence of complications. *Conclusions:* This study highlights the need for personalized treatment approaches and tailored cardioprotective strategies to improve outcomes and enhance the quality of life for oncology patients. Future studies should prioritize developing improved strategies to reduce the cardiovascular complications linked to contemporary cancer treatments.

## 1. Introduction

Cancer survivors encounter various challenges, including the potential for cancer recurrence and heart-related complications. Cardio-oncology emphasizes the importance of a collaborative, interdisciplinary approach to patient care, facilitating the effective management of cancer patients experiencing cardiovascular challenges [1,2]. Research in this field remains in its early stages and faces certain limitations. The outcomes observed in various studies often differ based on factors such as age, gender, ethnicity, marital status, cancer stage, time of diagnosis, and type of surgery. As a result, further comprehensive research is necessary to validate these findings [3]. Furthermore, numerous aspects of the interplay between cardiovascular and anticancer medications, as well as the genetic factors influencing their mechanisms, remain insufficiently understood. Clinical trials are essential to establish the best timing for pausing or resuming cancer treatments in cases of cardiovascular complications; it is essential to investigate suitable alternative treatment options when necessary [4].

Tailored strategies are vital for patients with both cancer and heart disease to minimize therapy-related cardiac risks. Oncologists must manage cardiovascular complications, while cardiologists should be aware of cancer risks linked to certain heart medications [5,6].

The rising cardiotoxicity risk from advanced cancer treatments underscores the need for updated protocols based on current research. Cardio-oncology uses a multidisciplinary approach to deliver integrated care for patients with cancer and heart conditions. This specialized field is pivotal in enhancing patient outcomes by focusing on the prevention, detection, and management of cardiovascular issues linked to cancer treatments. The present study investigates the cardiotoxic impacts of anticancer therapies and explores innovative diagnostic and treatment strategies aimed at improving care for affected patients [7,8].

Over time, the fields of oncology and cardiology have become deeply interconnected. Cardiologists with expertise in cardio-oncology play a vital role in fostering collaboration with oncologists, driving progress in both specialties. The Cardio-Oncology Rehabilitation and Education (CORE) approach, modeled after the CR framework, offers significant potential to enhance outcomes for cancer survivors [2,9]. Healthcare policymakers should broaden cardiac rehabilitation (CR) programs to encompass cancer patients, with the goal of lowering mortality rates. Cardio-oncology must evolve beyond solely addressing cardiotoxicity, while oncologists should integrate effective strategies for preventing and managing cardiovascular diseases (CVD). These strategies should include appropriate medications and lifestyle interventions, such as structured exercise programs [10].

Studies show that cancer significantly contributes to noncardiac mortality in heart failure (HF) patients, who are also at an increased risk of developing cancer. Shared factors like aging, obesity, and smoking partly explain this link, but emerging evidence of a possible causal connection between HF and cancer highlights the need for deeper investigation. The systemic effects of both conditions and their interactions are increasingly evident. This emerging area of study has been termed “reverse cardio-oncology” [11,12].

A major challenge in breast cancer treatment is the cardiotoxicity linked to traditional therapies, especially HER2-targeted drugs like trastuzumab, which, despite their efficacy against HER2-positive breast cancer, can cause congestive heart failure (CHF) [13]. The combination of trastuzumab with anthracycline chemotherapy significantly heightens the risk of cardiac injury. Studies show that 8.7% of patients treated with trastuzumab experience significant declines in left ventricular ejection fraction (LVEF), potentially leading to heart failure [14,15].

Pertuzumab, another antibody used for HER2-positive breast cancer, has a similar but less severe cardiotoxicity profile. When combined with trastuzumab, it may amplify cardiac side effects; however, the therapeutic advantages for patients remain substantial [16]. Antibody–drug conjugates, such as trastuzumab emtansine (T-DM1), are effective in treating breast cancer and show reduced cardiotoxicity compared to traditional trastuzumab. Newer agents like trastuzumab deruxtecan (T-DXd) offer even lower cardiotoxicity but present additional risks, such as interstitial lung disease [17].

This study examines the cardiotoxic effects of cytostatic therapies in connection with cancer stage, histopathological features, and TNM classification. It seeks to identify relationships between tumor characteristics and the severity of cardiotoxicity, offering a deeper understanding of the cardiovascular risks linked to different cancer types and stages treated with cytostatic therapies. The study seeks to generate actionable data that will enhance therapeutic protocols through a personalized approach, integrating the tumor’s histopathological profile and TNM classification. This tailored strategy aims to prevent and manage cardiotoxicity more effectively, thereby improving the overall quality of life for cancer patients.

## 2. Materials and Methods

### 2.1. Study Design

This retrospective study evaluated the cardiotoxic effects of breast cancer therapies and their influence on patients’ quality of life (QOL). Conducted at Hospital Oncohelp in Arad, Romania, between January 2008 and December 2023, the study received ethical approval from the hospital’s Ethics Committee (approval code Nr.CEFMF 8/27 June 2024), with all participants providing written informed consent. The research followed the ethical standards of the Declaration of Helsinki and adhered to CONSORT guidelines for observational studies. The study flowchart can be seen in Figure 1.

### 2.2. Patient Population

A total of 5149 female patients, aged 20 to 88 years, with histologically confirmed breast cancer were enrolled in this study. Participants were selected based on the following inclusion criteria: a diagnosis of stage I-IV breast cancer. Subdivisions within cancer stages, such as 2A, 2B, 3A, and 3B, provide a more detailed understanding of disease progression compared to broader stages (1, 2, 3, 4). These distinctions reflect differences in tumor size, lymph node involvement, and local invasion, which are critical for tailoring treatment and predicting outcomes. For example, stage 2A typically involves smaller tumors with limited lymph node spread, while 2B includes larger tumors or greater nodal involvement. Similarly, 3A often represents significant lymphatic involvement, while 3B indicates local tissue invasion. These nuanced classifications enhance treatment precision and prognostic accuracy. In this study, subdivided stages were used.

Exclusion Criteria: Patients with a history of significant cardiovascular disease, as defined by the European Society of Cardiology (ESC) guidelines—including coronary artery disease, congestive heart failure, or prior myocardial infarction—were excluded from the study. Additional exclusions applied to individuals with conditions that hindered their ability to complete study evaluations or those unable to provide informed consent.

The study population included patients receiving standard chemotherapy regimens or HER2-targeted therapies, enabling an analysis of diverse treatment approaches. The most frequently administered agent was Herceptin (trastuzumab, Roche Registration GmbH, Germany), used either alone or in combination with other therapies. In some cases, trastuzumab was paired with Bondronat (ibandronic acid, Roche Pharma AG, Germany) and tamoxifen to enhance treatment efficacy. Kadcyla (trastuzumab emtansine, Roche Pharma AG, Germany) was another common choice, often used alone or with bisphosphonates such as ibandronic acid for patients requiring additional bone support.

Other treatment regimens included combinations like paclitaxel (TAX) with trastuzumab and sequential administration of docetaxel and cyclophosphamide with trastuzumab for advanced-stage patients. In some instances, trastuzumab emtansine was administered in 21-day cycles, with dosage adjustments (up to 80%) based on individual tolerance levels. These regimens highlight the tailored therapeutic strategies employed to address the specific needs of breast cancer patients. These personalized treatment strategies aimed to maximize therapeutic efficacy while mitigating the risks of cardiotoxicity associated with HER2-targeted therapies and chemotherapy.

Additional inclusion criteria required adequate organ function and the ability to complete study questionnaires.

### 2.3. Breast Cancer Treatment and Cardiotoxicity Monitoring

Patients were treated according to established clinical protocols, with cardiotoxicity closely monitored due to the well-documented cardiac effects of certain chemotherapy agents, such as anthracyclines and trastuzumab. Cardiac monitoring involved echocardiographic assessment (Philips Affiniti CVx, Netherlands) of left ventricular ejection fraction (LVEF) at three critical stages: before starting treatment, midway through therapy, and after its completion. Cardiotoxicity was defined in accordance with the Common Terminology Criteria for Adverse Events (CTCAE), version 26, as a reduction in LVEF by 10% or more from baseline to a value below 50%.

### 2.4. Quality of Life Assessment

The performance index in this study is a measure used to evaluate the functional status and physical performance of patients, often reflecting their ability to perform daily activities and their overall fitness for treatment. The index is derived from clinical evaluations and follows a range from 0 to 2:A score of 0 indicates no limitations in daily activities or physical function;A score of 1 indicates mild limitations but the patient remains largely independent;A score of 2 indicates significant limitations, requiring substantial assistance or affecting treatment feasibility.

The primary measure, quality of life, was assessed using the EORTC QLQ-C30 (European Organization of Research and Treatment for Cancer Quality of Questionnaire—Core 30), along with the breast cancer-focused module QLQ-BR23. The EORTC QLQ-C30 reviews five functional areas (physical, role, emotional, cognitive, and social activity), three symptom categories (fatigue, nausea/vomiting, and discomfort), and a comprehensive global health scale [18].

#### 2.4.1. Questionnaire Administration:

Patients completed the questionnaires at three key time points:Baseline: Before the initiation of cancer treatment;Mid-treatment: After three cycles of chemotherapy;End of Treatment: Within 6–8 weeks following the final dose of therapy.

Questionnaires were delivered electronically on a secure system or as paper versions, depending on patient preference. Assistance was offered when necessary to confirm that responses were correctly documented.

#### 2.4.2. Scoring and Explanation:

Responses were evaluated based on the EORTC scoring system and adjusted into a 0–100 range:For functional sections and global health status, increased scores indicated improved functioning and quality of life.For symptom domains, elevated scores reflected heightened symptom intensity or more significant challenges.

This approach allowed for a comprehensive assessment of the patients’ quality of life throughout their cancer treatment journey.

### 2.5. Statistical Analysis

Data review was performed using SPSS, version 20 [19]. Statistical summaries were used to describe demographic and medical characteristics of the study participants. Variations in quality of life (QOL) scores over treatment were assessed using suitable statistical approaches, including paired *t*-tests, ANOVA, or Wilcoxon signed-rank comparisons, based on the distribution of collected data. The association between cardiotoxicity and QOL outcomes was investigated using Pearson correlation methods.

To handle incomplete data, a complete-case analysis approach was implemented. A *p*-value below 0.05 was treated as statistically meaningful for every analysis.

### 2.6. Data Availability

All relevant data used in this research, including anonymized patient details and raw scores, have been stored in a publicly available repository. Any analytical code utilized for the study is accessible upon request, and all materials related to the research will be made available to readers upon reasonable inquiry.

### 2.7. Ethical Considerations

This research complied with the ethical standards set forth in the Declaration of Helsinki. Ethical approval for studies involving human subjects was granted by the Ethics Committee of the University of Oradea (approval code: CEFMF 8/26 June 2024). Written informed consent was obtained from all participants, and all data were anonymized prior to analysis to ensure patient confidentiality.

### 2.8. Protocols and Materials

All procedures for data collection, cardiotoxicity assessment, and quality-of-life (QOL) evaluation followed recognized guidelines. The application of the EORTC QLQ-C30 and QLQ-BR23 questionnaires is well-established and appropriately cited. Furthermore, this study introduced innovative approaches, including a tailored analysis of allopathic treatment durations to evaluate quality of life in the context of cardiotoxic effects.

## 3. Results

### 3.1. Demographic Description

The study analyzed data from a total of 7674 participants, divided into two groups: the exclusion group (2525 individuals) and the study group (5149 individuals). The mean age for the total cohort was 58.8 years (SD ± 12.94), with the exclusion group having a higher mean age of 61.48 years (SD ± 13.48) compared to the study group at 57.48 years (SD ± 12.51). Average weight across all participants was 70.36 kg (SD ± 15.92), with slightly higher values in the exclusion group (71.77 kg, SD ± 17.79) compared to the study group (69.67 kg, SD ± 14.41). Mean height was 161.18 cm (SD ± 7.12), with the exclusion group averaging 160.19 cm (SD ± 6.97) and the study group 161.66 cm (SD ± 7.51). Body surface area was similar across groups, averaging 1.72 m^2^ (SD ± 0.20) for the total cohort, with 1.71 m^2^ (SD ± 0.21) in the exclusion group and 1.73 m^2^ (SD ± 0.18) in the study group. The performance index was consistent at 0.60 (SD ± 0.72) for the total group, with minor variations between the exclusion group (0.61, SD ± 0.069) and the study group (0.60, SD ± 0.075).

Hypertension was reported in 19.80% of the total cohort, with a higher prevalence in the exclusion group (21.11%) compared to the study group (19.17%). Diabetes was observed in 6.02% of participants, more frequently in the exclusion group (7.68%) than in the study group (5.21%). Chronic liver diseases affected 7.67% of the total cohort, chronic kidney diseases 2.24%, and chronic gastrointestinal diseases 12.04%, with the exclusion group consistently showing higher prevalence rates for these conditions. Chronic gastrointestinal diseases were particularly prominent in the exclusion group (24.78%) compared to the study group (5.79%). This analysis provides a comprehensive overview of the demographic and clinical characteristics of the participants (Figure 2).

Figure 3 illustrates a rising trend in new cases from 2009 to 2016, reaching a peak of 370, followed by stabilization with minor fluctuations, culminating at 288 in 2023. The number of deaths showed variability, peaking at 230 in 2018 before gradually decreasing to 170 in 2022, with a slight increase to 197 in 2023. The increase in cases alongside stable or declining death rates may indicate advancements in treatment, earlier detection, or improved patient management, contributing to enhanced survival rates over time.

Initially, 7674 patients were evaluated (Table 1). From these, 32.91% (2525 individuals with an average age of 61.31 years, SD: 14.87) presented significant pre-existing cardiovascular conditions and were omitted from further analysis. The main focus of the research was to examine the cardiotoxic impacts of oncology treatments, leading to a finalized cohort of 5149 participants with a mean age of 57.48 years (SD: 12.51).

This study analyzes the distribution of LVEF values in a cohort of 5149 participants (Figure 4). The majority (96%) had LVEF values between 50% and 65%, while 4% recorded values below 50%. The cumulative data confirms that all individuals were categorized within these ranges. Regarding heart problems, 50.5% of participants reported experiencing them, while 49.5% did not. Analysis of the performance index revealed that 57.3% of the sample had a score of 1.00, 41% had a score of 0.00, and a small subset (1.6%) achieved a score of 2.00.

### 3.2. Differential Diagnosis

In this section, symptoms, disease progression, and treatments were selected to illustrate the diversity of the study population, highlighting variations in symptoms, disease progression, and treatment regimens to provide context for the findings on cardiotoxicity and its relationship to clinical and therapeutic factors.

In our study, we encountered various cases of breast neoplasms, each presenting distinct characteristics and treatment approaches. One case involved a left breast neoplasm, treated with a modified radical procedure and left axillary lymphadenectomy during October 2014, followed by four rounds of adriamycin (doxorubicin) and cyclophosphamide in combination chemotherapy conducted between November 2014 and January 2015. The individual additionally had hypertension and type 2 diabetes, alongside a history of total thyroidectomy due to a multinodular goiter in October 2013. This specific case represented 8% of the overall cases. The allocation of breast cancer instances by location and classification appears in Figure 5.

Another patient with a right breast neoplasm received six courses of CMF and herceptin from June to November 2014, followed by a mastectomy and right axillary node removal in December 2014. This case comprised 13% of the total cases.

A third case involved a right breast neoplasm with carcinomatous mastitis. The patient underwent a biopsy and received six courses of epirubicin and taxane combined chemotherapy between November 2013 and March 2014, followed by irradiation with a cumulative dose of 50 Gy during May to June 2014. The individual also had grade II hypertension, chronic ischemic conditions, and grade II obesity. This instance represented 4% of the overall cases.

Another left breast tumor was managed with a Madden mastectomy and axillary lymphadenectomy in August 2013, accompanied by a rheumatoid arthritis diagnosis, comprising 9% of total cases.

In one situation, a left breast tumor was treated using a sectorectomy and left axillary lymphadenectomy on 28 October 2014, while concurrently addressing hypertension and type 2 diabetes. This accounted for 2% of the cases.

In another instance, a left breast tumor received four courses of EC, four cycles of docetaxel, and herceptin therapy between February and July 2014, along with 50 Gy of radiation to the left breast, axillary, and supraclavicular nodes in August and September 2014. This contributed 11% to the case distribution.

A more severe case included a left breast neoplasm with pleural, lymphatic, and liver metastases, representing 6% of cases.

One patient with a left breast tumor underwent a radical mastectomy alongside six cycles of EC chemotherapy. Currently receiving hormone therapy, the individual also experienced a left femoral neck dislocation and mild secondary anemia. This comprised 2.3% of all instances.

Another complex scenario involved a left breast neoplasm with metastases affecting vertebral, costal, and scapular bones, bilateral lungs, and supraclavicular nodes. The individual underwent a radical mastectomy and left axillary lymphadenectomy in July 2012, followed by eight cycles of EC chemotherapy spanning August 2012 to January 2013. This accounted for 5.2% of cases.

Lastly, a left breast tumor was managed with a single EC chemotherapy course in June 2014, a sectorial mastectomy in July 2014, and six cycles of docetaxel combined with herceptin from August to December 2014. Total radiation doses of 48 Gy and 50 Gy targeted distinct areas of the left breast and axillary nodes in February 2015. This case represented 1.8% of the total.

### 3.3. Histopathology

Histopathological evaluation identified diverse features of infiltrative ductal carcinoma and other breast cancer subtypes, emphasizing significant variability in tumor type and aggressiveness.

One case revealed infiltrative ductal carcinoma with perineural invasion and lymphatic emboli. The immunohistochemical (IHC) profile showed estrogen receptor (ER) expression at 95%, progesterone receptor (PR) at 98%, a Ki67 index of 30%, and a HER2/neu score of +3, indicating high proliferative potential, responsiveness to hormonal therapies, and increased aggressiveness due to HER2 positivity.

In January 2012, cytological examination confirmed infiltrative ductal carcinoma, with residual invasive carcinoma detected by July 2012, including perineural invasion and metastases in 6 of 12 lymph nodes. The IHC profile revealed ER at 20%, PR at 90%, Ki67 at 30%, and HER2 scored as 2+, pending FISH testing for gene amplification.

In January 2013, a grade G3 invasive ductal carcinoma was identified, involving 3 of 13 lymph nodes. The IHC profile indicated HER2 at 3+, Ki67 at 90%, and no expression of ER or PR. By September 2013, lymph nodal metastases were confirmed, with biopsies showing negative ER and PR, a Ki67 index of 30%, and HER2 at 3+.

Another case involved G2 invasive ductal carcinoma without lymphatic or vascular invasion but with metastases in 2 of 17 lymph nodes. The IHC profile showed ER at 90%, PR at 0%, Ki67 at 10%, and HER2 at 2+, with gene amplification confirmed by CISH.

Other cases included invasive ductal carcinoma with trabecular, glandular, and solid growth patterns, sometimes coexisting with invasive lobular carcinoma. The IHC profile revealed ER at 90%, PR at 90%, Ki67 at 30%, and HER2 at 2+. In October 2013, biopsies identified carcinomatous emboli in lymphatic vessels, with ER and PR at 0%, unspecified Ki67, and HER2 at 2+, with gene amplification confirmed.

Additional findings included poorly differentiated, high-grade ductal carcinoma with ER at 90% and a Ki67 index of 80%. Another case involved a G3 invasive ductal carcinoma with advanced features with metastases in 1 of 9 lymph nodes and an IHC profile showing ER at 60%, PR at 75%, and HER2 at 2+, with confirmed gene amplification.

Overall, the histopathological analysis revealed a broad spectrum of breast tumor types, varying significantly in immunohistochemical profiles and levels of aggressiveness. These findings underscore the necessity of tailoring treatment strategies to the unique characteristics of each tumor, presented in Table 2.

### 3.4. TNM and the Risk of Cardiac Dysfunction in Patients with LVEF < 50%

The TNM classification (Table 3) is a pivotal framework in oncology for determining cancer stage, focusing on three essential elements: T, signifying the size and extent of the primary tumor; N, denoting lymph node involvement; and M, representing distant metastases. This classification provides a comprehensive understanding of cancer progression and assists in designing tailored treatment approaches. Based on collected data, the TNM stages are categorized as follows.

Advanced stages with significant involvement: pT4b Nx M1 (hep, lung, skin)—an advanced tumor (T4b) with considerable local extension, potentially affecting adjacent structures. Lymph node status is unspecified (Nx), with metastases in the liver, lungs, and skin; T4c N2 M1 (oss, pul, lym)—a primary tumor (T4c) showing substantial local growth, involving regional lymph nodes (N2) and metastases in bones, lungs, and lymph nodes; T4d Nx M1 (oss, hep, pul, lym)—a highly advanced tumor (T4d) with metastases in bones, liver, lungs, and lymph nodes, with undefined lymph node status (Nx).

Advanced stages with nodal involvement and metastases: pT2 N1 M1 (oss, pul, lym)—a moderately sized tumor (T2) with regional lymph node involvement (N1) and metastases in bones, lungs, and lymph nodes; pT1 N1 M1 (pul)—a smaller tumor (T1) affecting a lymph node (N1) with metastases confined to the lungs (M1); cT4b N1 M0—an advanced tumor (T4b) involving lymph nodes (N1) but no distant metastases (M0).

Intermediate and localized stages: T3 N1 M0, ypT2 N1 M0—a larger tumor (T3) involving lymph nodes (N1) without distant metastases (M0). Post-treatment reassessment indicated reduced tumor size (ypT2 N1 M0) with no distant spread; T2 N0 M0—a moderately sized tumor (T2) without lymph node involvement (N0) or distant metastases (M0); T1b N1 M0, ypT1c N1a M0—a smaller tumor (T1b) with lymph node involvement (N1) and no metastases (M0). Following treatment, the stage adjusted to ypT1c N1a M0, showing slight tumor modifications without metastases.

Stages with limited invasion and no metastases: pT2 N0 M0—a moderately sized tumor (T2) without lymph node involvement (N0) or metastases (M0); T4d Nx M0—a larger tumor (T4d) with extensive local spread but without lymph node data (Nx) or distant metastases (M0); pT1c N0 M0—a smaller tumor (T1c) with no lymph node involvement (N0) or metastases (M0).

Recurrences and metastases: pT1 N1 M1 (pul)—a smaller tumor (T1) involving a lymph node (N1) with lung metastases (M1); cT3 N1 M1 (oss), ypT2 N0 Mx—a moderately sized tumor (T3) with lymph node involvement (N1) and bone metastases (M1). After treatment, the stage changed to ypT2 N0 Mx, reflecting tumor reduction and incomplete distant metastasis data (Mx).

Among the 207 patients with left ventricular ejection fraction (LVEF, FEVS) < 50%, the majority (107 patients, 51.7%) were in the advanced tumor stage (T4). This trend may reflect a higher prevalence of significant cardiac dysfunction in advanced cancer stages, potentially influenced by both the intensity of cancer treatments and the physiological burden of disease progression. Patients with T3 tumors comprised 54 cases (26.1%), while those with early-stage tumors (T1 and T2) represented smaller proportions, with 29 (14.0%) and 17 (8.2%) patients, respectively. Although this study cannot definitively distinguish whether the reduced LVEF is primarily due to treatment effects or tumor progression, these findings underscore the critical need for close cardiac monitoring in patients with advanced TNM stages to address potential cardiotoxicity.

### 3.5. Cardiotoxicity

Congestive heart failure represents a significant adverse consequence of trastuzumab therapy, highlighted by a decrease in left ventricular ejection fraction (LVEF), especially in older individuals and those with existing health factors at greater risk. Fortunately, this decline in LVEF is reversible in most cases. Current guidelines advocate for routine LVEF monitoring and the implementation of cardioprotective strategies to prevent or address cardiac dysfunction. Importantly, HER2-targeted therapy can often be continued in patients with mild to moderate cardiac impairment.

Figure 6 summarizes the case processing of LVEF across different stages at detection and age groups. It shows that for LVEF < 50%, there were 105 cases at stage 1.00, 34 at stage 2.00, and smaller numbers across higher stages, with all cases being valid (no missing data). For LVEF between 50 and 65%, there were significantly more cases, ranging from 507 at stage 1.00 to 2055 at stage 4.00, again with no missing data. This highlights the distribution of cases by stage and LVEF category, indicating a higher frequency of cases in the 50–65% LVEF range, particularly at advanced stages.

Figure 7 provides an overview of case processing for LVEF across various stages at detection, categorized by age, weight, height, and performance index. For LVEF < 50%, cases are relatively fewer, with 105 at stage 1.00 and declining numbers at higher stages, all with no missing data. In contrast, for LVEF between 50 and 65%, the number of cases is significantly higher, ranging from 507 at stage 1.00 to 2055 at stage 4.00, consistently showing no missing data. Across all categories, the data indicates a pronounced concentration of cases in the 50–65% LVEF range, especially at advanced stages.

Table 4 provides Pearson correlation coefficients for various health and performance variables, including LVEF, heart problems, age, weight, height, body surface area, and performance index, using data from 5149 participants. Significant negative correlations are observed for LVEF with heart problems (−0.153), weight (−0.359), height (−0.078), and body surface area (−0.308), all at the 0.01 significance level. Positive correlations are noted between heart problems and weight (0.597) as well as body surface area (0.562). Age shows weak but significant positive associations with heart problems (0.138) and weight (0.140), while being negatively correlated with height (−0.205). The strongest correlation is between body surface area and weight (0.938). The performance index displays weak but significant correlations with age (0.247) and heart problems (0.096). Most relationships are statistically significant, with only a few exceptions.

### 3.6. Quality of Life

The data analyzes quality of life (QoL) scores, assessed by the EORTC QLQ-C30, across three left ventricular ejection fraction (LVEF) categories: <50%, 50–65%, and >65%. Patients with LVEF below 50% reported a mean QoL score of 63.17 (SD ± 7.80), suggesting that reduced heart function may slightly affect QoL, though scores remain fairly stable. For patients within the normal LVEF range (50–65%), the mean score was 64.28 (SD ± 8.39), showing slightly higher results and greater variability. Data for LVEF > 65% was unavailable, restricting further comparison. These findings indicate that QoL remains similar overall for patients with reduced and normal heart function, with some individual differences observed.

A non-parametric analysis investigated the relationship between LVEF and QoL scores. The Mann–Whitney U test revealed a U value of 466,556.000, a Wilcoxon W value of 488,084.000, and a Z-score of −2.147, with an asymptotic significance (2-tailed) of 0.032, indicating a statistically significant difference in QoL scores between LVEF categories (*p* < 0.05). Specifically, patients with lower LVEF (<50%) experienced a modest but significant reduction in QoL compared to those with normal LVEF, highlighting the importance of monitoring cardiotoxicity during breast cancer therapy.

LVEF served as the grouping variable to evaluate differences between predefined categories (<50%, 50–65%). Figure 8A illustrates the impact of LVEF on QoL, while Figure 7B highlights that stage I patients with low LVEF experienced the greatest reduction in QoL, followed by those in stage 3A. These results emphasize the importance of addressing cardiotoxicity to maintain quality of life during cancer care.

Figure 8B summarizes the case processing of EORTC QLQ-C30 quality of life scores across different LVEF (left ventricular ejection fraction) categories and stages at detection, highlighting the number of valid and missing cases for each combination. For LVEF < 50%, cases are distributed across stages 1.00 to 4.00, with totals ranging from 11 (stage 2.50) to 105 (stage 1.00), and all cases are valid with no missing data. In the 50–65% LVEF range, the number of cases is significantly higher, ranging from 507 (stage 1.00) to 2055 (stage 4.00), also with no missing data. This indicates a higher concentration of cases in the 50–65% LVEF range, particularly in advanced stages, with complete data availability for analysis. The figure reflects a focus on patients with higher LVEF values, emphasizing their predominance in the dataset.

## 4. Discussion

Recent advances in cancer therapies have markedly improved patient survival rates, yet they have also introduced significant risks to cardiovascular [20] and cardiometabolic health [21]. Cardiotoxicity associated with chemotherapy, radiation, and targeted therapies has become a critical concern. Supporting our findings, numerous studies highlight the necessity of cardiac monitoring during anticancer treatments, particularly with anthracyclines, trastuzumab, and tyrosine kinase inhibitors [22,23].

Anthracyclines are well-documented in the literature for causing cardiomyocyte necrosis and fibrosis, a severe, dose-limiting complication [24]. These observations stress the importance of identifying high-risk patients early to mitigate cardiac dysfunction from these treatments.

Radiation therapy, especially to the mediastinal or thoracic regions, further exacerbates cardiotoxic risk by accelerating atherosclerosis, myocardial infarction, and heart failure, underscoring the need for continuous cardiac evaluation [25]. Our study confirms the value of echocardiographic tools, such as global longitudinal strain and left ventricular ejection fraction (LVEF), for detecting systolic dysfunction. This aligns with studies recommending 3D echocardiography as a reliable monitoring approach during chemotherapy [26,27].

In the biomarker domain, our research emphasizes the utility of cardiac troponins and natriuretic peptides (BNP and NT-proBNP) for early cardiotoxicity detection [28,29]. These markers can reveal subclinical cardiac dysfunction prior to symptom onset, corroborating existing evidence [30]. Additionally, emerging biomarkers like myeloperoxidase and microRNAs offer promising avenues for the early detection of cardiac injury [31].

Scientific literature underscores the critical need for collaboration between oncologists and cardiologists, emphasizing comprehensive cardiological assessments before, during, and after cancer treatment [32]. Additionally, personalized management strategies for high-risk patients are strongly advocated to mitigate cardiotoxicity while maintaining the effectiveness of cancer therapies [33].

Radiotherapy has been associated with coronary heart disease and fibrotic alterations in the heart valves, pericardium, and myocardium [34,35]. To reduce the risk of complications, patients undergoing chemotherapy—particularly those with pre-existing cardiovascular conditions—should receive thorough cardiovascular evaluations prior to treatment. Monitoring left ventricular systolic function and cardiac biomarkers has proven especially valuable for high-risk individuals [36].

Cardioprotective interventions, such as ACE inhibitors, angiotensin receptor blockers, and beta-blockers, have demonstrated success in minimizing chemotherapy-induced cardiotoxicity. Moreover, antiplatelet or anticoagulation therapies may benefit patients with an elevated risk of hypercoagulability associated with cancer or chemotherapy regimens [37,38].

Collaboration between cardiologists and oncologists is essential for effectively managing the cardiovascular risks associated with cancer treatments. Advanced therapies, including radiation, cytotoxic chemotherapy, targeted molecular inhibitors, and immune-modulating antibodies, pose significant challenges to cardiovascular health [39]. While the toxicities of traditional radiotherapy and chemotherapy are well-documented, there is limited data on the long-term effects of newer molecular therapies and immunotherapies, raising concerns about their potential to impair the quality of life and survival of cancer patients, particularly those with pre-existing cardiovascular conditions [40].

Cardiovascular complications from cancer treatments, such as hypertension, venous thromboembolism, coronary heart disease, valvular disease, heart failure, and arrhythmias, highlight the need for early detection of subclinical cardiotoxicity. This underscores the importance of a multidisciplinary cardio-oncology approach for prevention, diagnosis, and treatment [41,42]. This study explores the cardiotoxic effects of various treatments and investigates novel diagnostic and therapeutic strategies to improve patient outcomes.

Elderly patients, those with existing cardiovascular risk factors, or those with a history of chemotherapy or radiotherapy are particularly vulnerable to cardiotoxicity [21,43]. Routine monitoring through troponins, echocardiography, or cardiovascular magnetic resonance can detect cardiac issues before the progression to heart failure [44]. The increasing importance of serum biomarkers in risk assessment and monitoring is also supported by recommendations from the European Society of Cardiology [45].

The EORTC QLQ-C30 is a widely recognized instrument for assessing quality of life in breast cancer patients, measuring the effects of disease progression and treatment-related side effects on overall well-being [46,47,48]. Studies indicate that patients in advanced cancer stages or those experiencing cardiotoxicity, particularly from trastuzumab, often report lower scores in functional domains such as physical, role, and emotional functioning [13,49]. Additionally, reduced left ventricular ejection fraction (LVEF) has been linked to greater fatigue and decreased physical performance, reflecting the adverse cardiac effects of treatment. Consistent with these findings, our study observed slightly lower EORTC QLQ-C30 scores in patients with LVEF < 50% compared to those with normal LVEF (50–65%), emphasizing the impact of cardiotoxicity on quality of life. While previous research often associates more significant quality of life impairments with advanced cancer stages, our results demonstrate that even stage I patients with low LVEF experience notable quality of life reductions, underscoring the importance of early cardiac monitoring and timely intervention.

### Limitations

This study highlights the importance of cardio-oncology and regular cardiovascular monitoring during cancer treatment. However, it is limited by the absence of long-term data on newer molecularly targeted therapies and immunotherapies. Further research is necessary to explore late-stage cardiovascular effects and their influence on quality of life in cancer survivors. Despite these limitations, the findings underscore the urgent need for personalized treatment strategies that reduce cardiotoxicity and improve outcomes for oncology patients.

## 5. Conclusions

Congestive heart failure is a notable adverse effect of trastuzumab therapy, especially in elderly patients or those with existing cardiovascular risk factors, typically manifesting as a reversible reduction in LVEF. Routine monitoring of LVEF and the implementation of cardioprotective measures enable the continuation of HER2-targeted treatments in patients with mild to moderate dysfunction, ensuring treatment effectiveness is maintained. Our findings reveal that stage I breast cancer patients with low LVEF experience the most notable quality of life reductions, highlighting the compounded impact of cardiotoxicity and cancer progression. The distribution of patients with LVEF < 50% across TNM stages highlights a significant association between advanced tumor stages (T4) and increased risk of cardiac dysfunction, emphasizing the need for enhanced cardiac monitoring and tailored cardioprotective strategies in patients undergoing treatment for advanced breast cancer. These results underscore the importance of early cardiac screening, multidisciplinary care, and personalized management to optimize outcomes and maintain quality of life during breast cancer treatment.

## Figures and Tables

**Figure 1 medicina-61-00301-f001:**
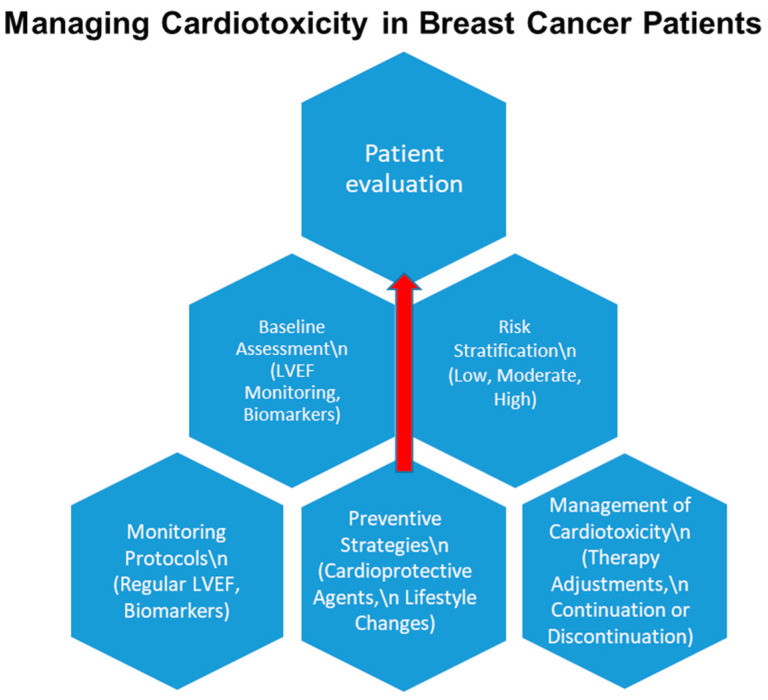
Study flowchart.

**Figure 2 medicina-61-00301-f002:**
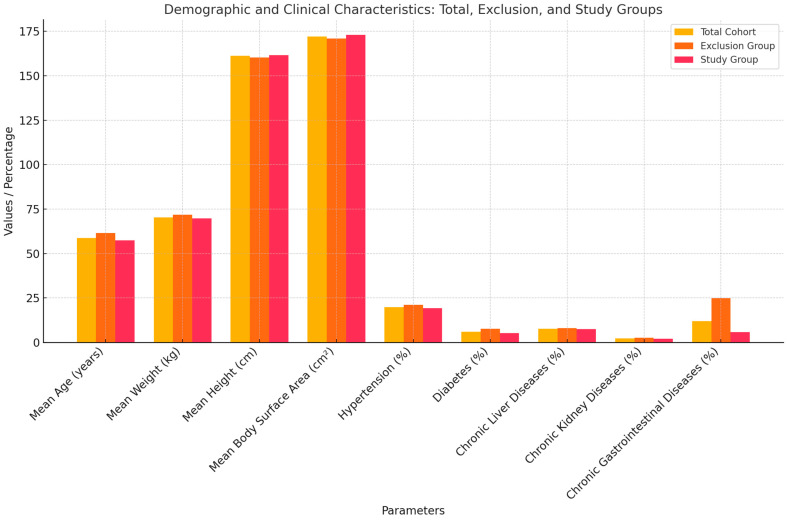
The demographic profile of the study cohort includes details such as age, weight (in kilograms), height (in centimeters), body surface area (in cm^2^), and the prevalence of conditions like hypertension, diabetes, and chronic liver, kidney, and gastrointestinal diseases.

**Figure 3 medicina-61-00301-f003:**
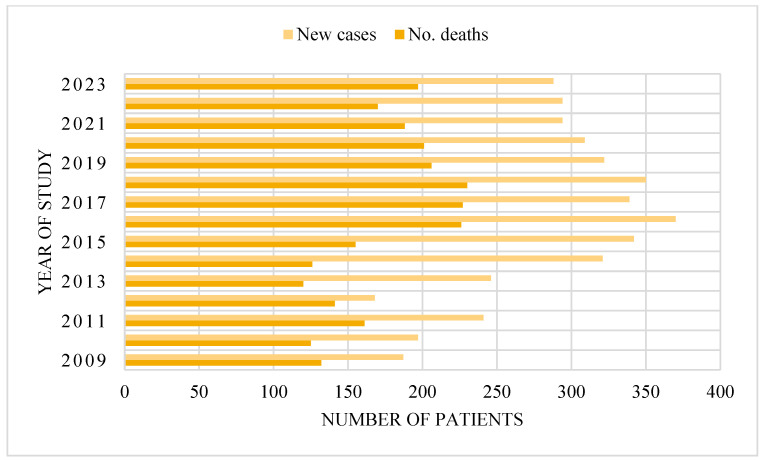
Trends in new cases and mortality from 2009 to 2023.

**Figure 4 medicina-61-00301-f004:**
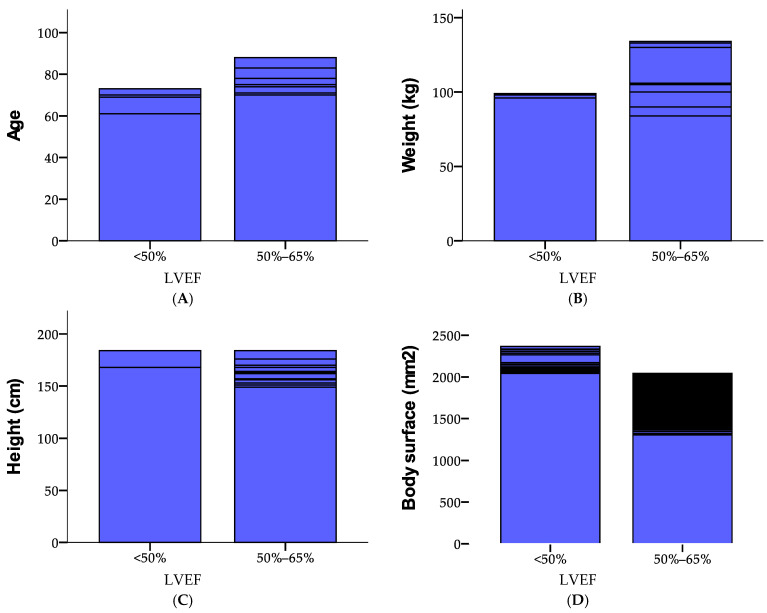
Demographic attributes of the study cohort, including age (**A**), weight in kilograms (**B**), height in centimeters (**C**), and body surface area in kg/mm^2^ (**D**), depending on the category of FEVS. The blue bars represent the average or mean values and the black lines indicate variability in the data, such as standard deviation, range, or quartiles, showing the spread or differences depending on each case.

**Figure 5 medicina-61-00301-f005:**
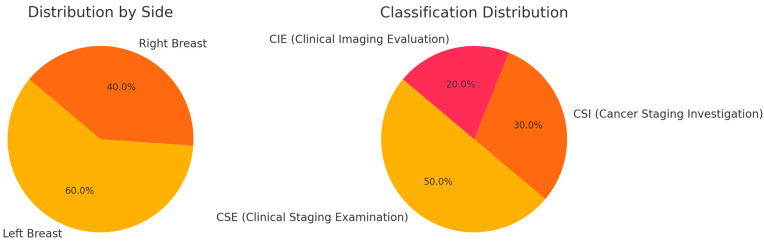
Distribution of breast cancer cases by side and classification.

**Figure 6 medicina-61-00301-f006:**
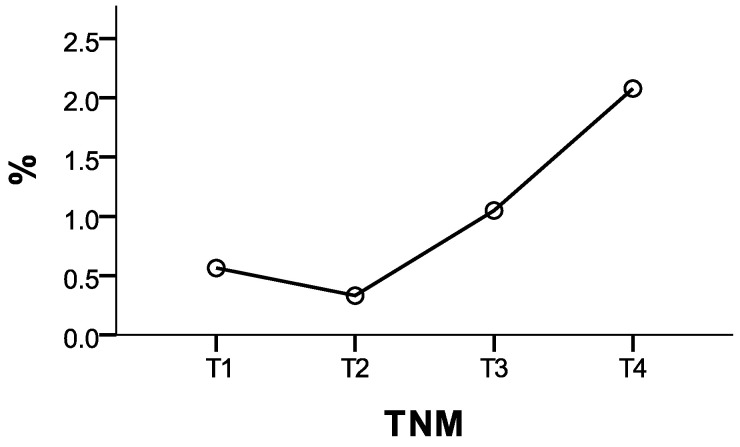
Distribution of patients with LVEF < 50% across TNM stages.

**Figure 7 medicina-61-00301-f007:**
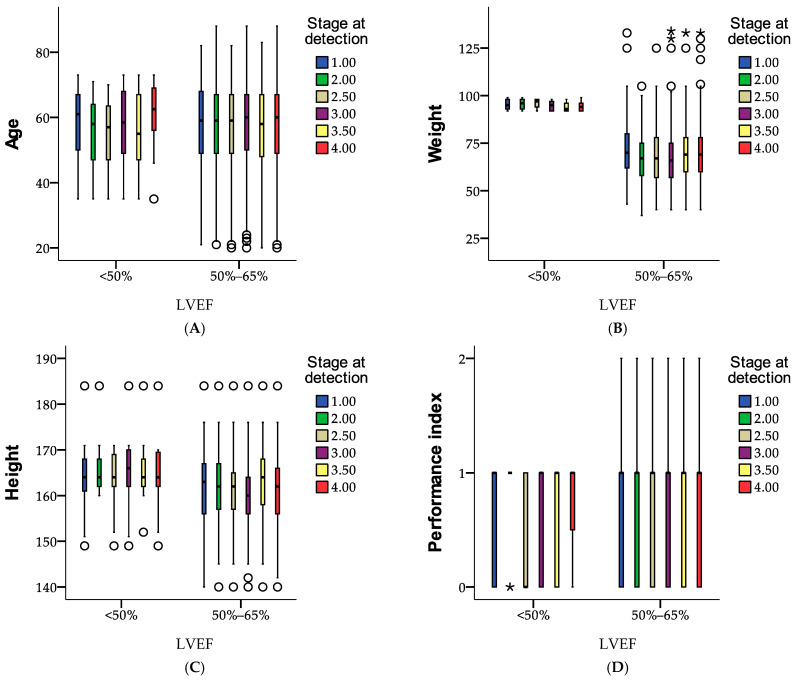
Examination of health metrics by LVEF groups for age (**A**), weight (**B**), height (**C**), and performance levels (**D**). ○ = exceptional cases; * = isolated cases.

**Figure 8 medicina-61-00301-f008:**
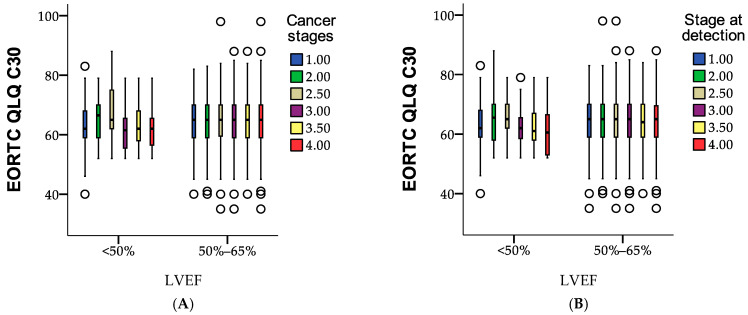
Boxplot representation of EORTC QLQ-C30 scores across LVEF categories and stages at detection: case processing summary. The figure visualizes (**A**) the distribution of EORTC QLQ-C30 scores based on left ventricular ejection fraction (LVEF) categories (<50% and 50–65%) and (**B**) stages of cancer at the time of detection, using a boxplot to illustrate data comprehensively. ○ = exceptional cases.

**Table 1 medicina-61-00301-t001:** Summary statistics of health and performance metrics for a cohort of 5149 participants.

Variable	Mean	Std. Deviation	Skewness	Kurtosis	Minimum	Maximum
LVEF	0.95	0.19	−4.68	19.93	0.00	1.00
Heart Problems	0.50	0.50	−0.02	−2.00	0.00	1.00
Age	57.48	12.50	−0.50	−0.14	20.00	88.00
Weight	69.67	14.41	0.54	0.64	37.00	134.00
Height	161.66	7.50	0.05	0.64	140.00	184.00
Body Surface Area	1.73	0.18	0.10	−0.20	1.303	2.365
Performance Index	0.60	0.52	−0.08	−1.22	0.00	2.00

**Table 2 medicina-61-00301-t002:** Immunohistochemical and pathological characteristics of invasive ductal carcinoma.

Tumor Type	Grade	Nodal Invasion	ER (%)	PR (%)	Ki67 (%)	HER2	FISH/CISH	Metastasis
Infiltrative Ductal Carcinoma	-	-	95	98	30	+3	-	None
Infiltrative Ductal Carcinoma	-	6/12	20	90	30	2+	CISH (+)	None
Invasive Ductal Carcinoma	G3	3/13	0	0	90	3+	-	Lymph nodular metastases
Invasive Ductal Carcinoma	G2	2/17	90	0	10	2+	CISH (+)	None
Invasive Ductal and Lobular Carcinoma	-	-	90	90	30	+2	-	None
Infiltrative Ductal Carcinoma	-	-	0	0	Not Specified	2+	CISH (+)	Carcinomatous emboli
Poorly Differentiated Ductal Carcinoma	-	-	90	-	80	-	-	None
Invasive Ductal Carcinoma	G3	1/9	60	75	30	2+	CISH (+)	None

IDC = invasive ductal carcinoma; G2/G3 = tumor grade; ER = estrogen receptor; PR = progesterone receptor; Ki67 = cell proliferation marker; HER2 = human epidermal growth factor receptor 2; FISH = fluorescence in situ hybridization; CISH = chromogenic in situ hybridization.

**Table 3 medicina-61-00301-t003:** TNM classification of breast cancer.

Attribute	Case 1	Case 2	Case 3	Case 4	Case 5	Case 6
T (Tumor)	T1b	T2	T3	T4b	T4c	T4d
N (Nodes)	N1	N0	N2	N1	N2	N3
M (Metastasis)	M0	M1 (pul)	M1 (oss, pul, lym)	M1 (pul)	M1 (oss, hep, lym)	M1 (oss, hep, pul, lym)
Additional Details	Small tumor, 1 lymph node, no metastases	Moderate tumor, no lymph node involvement, lung metastases	Large tumor, extensive nodal involvement, bone, lung, and lymph node metastases	Tumor with significant local extension, involvement of 1 ganglion, lung metastases	Tumor with extensive local ex-tension, extensive nodal involvement, bone, liver, and lymph node metastases	Tumor with extreme local extension, very extensive nodal involvement, multiple metastases in bones, liver, lungs, and lymph nodes

**Table 4 medicina-61-00301-t004:** Pearson correlation of performance and health variables.

Correlations		LVEF	Heart problems
Age	r	−0.007	0.138 **
	*p*	0.604	0.000
Weight	r	−0.359 **	0.597 **
	*p*	0.000	0.000
Height	r	−0.078 **	0.203 **
	*p*	0.000	0.000
Body surface area	r	−0.308 **	0.562 **
	*p*	0.000	0.000
Performance index	r	−0.009	0.096 **
	*p*	0.534	0.000
	N	5149	

r = Pearson correlation coefficient; *p* = statistical significance; ** = correlation is significant at the 0.01 level (two-tailed).

## Data Availability

All the data processed in this article are part of the research for a doctoral thesis and are archived in the esthetic medical office, where the interventions were performed.

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
