# Peer review of "Association Between Advanced TNM Stages and Increased Risk of Cardiac Dysfunction in Patients with LVEF < 50%"

_medicina, 2025, doi:10.3390/medicina61020301_

Round 1

Reviewer 1 Report

Comments and Suggestions for Authors

This study analyzed demographic data and cardiac function data from cancer patients after anti-cancer treatment. It aims to investigate the association between advanced TNM Stages and risk of cardiac dysfunction in patients with LVEF <50%. This is important mitigate cardiovascular risks resulting from modern cancer therapies. However, author will benefit from addressing the following questions:

Major points:

1.       In the abstract, authors mentioned “Biomarkers, such as cardiac troponins and natriuretic peptides, were employed to detect subclinical cardiotoxicity.”. No data relevant to biomarker are shown in this paper.

2.       The paper aims to investigate the association between advanced TNM stages and risk of cardiac dysfunction in patients with LVEF <50%. None of the figure is showing such data. Instead the manuscript focused on age, weight, height and body surface and quality of life.

3.       Figure 3 did not show any LVEF values, but shows age, weight, height and body surface instead.

4.       Two figures are labeled as Figure 5.

5.       The content related to Vitamin D in the discussion section is not relevant to this manuscript.

Minor points:

1.       The demographic data shown from line 221 to line 240 are more clear to present in a graph.

2.       Pls define the exclusion group and the study group at the very beginning of the paragraph for the ease of reading.

3.       At line 280, what is AC short for?

4.       At line 291, what is ET short for?

5.       What does FEVS mean? Does it mean LVEF? If yes, pls unify the nomenclature through the entire manuscript.

6.       In figure 6, firstly, figure legend is unclear. A rephrasing is absolutely necessary. In addition, the label of cancer stage is confusing. What is stage 2A? Pls explain the meaning of each stage number shown in the figure. How does it corelate to TNM standards?

Author Response

Reviewer 1

We sincerely thank the reviewer for their valuable feedback and constructive suggestions, which have greatly contributed to improving the quality and clarity of our manuscript. Your insights have helped us refine key aspects of our study, and we appreciate the time and effort you dedicated to providing such thoughtful and thorough comments. We have carefully addressed all your recommendations and believe the revised manuscript is significantly enhanced as a result. Thank you for your invaluable guidance and support in this process.

This study analyzed demographic data and cardiac function data from cancer patients after anti-cancer treatment. It aims to investigate the association between advanced TNM Stages and risk of cardiac dysfunction in patients with LVEF <50%. This is important mitigate cardiovascular risks resulting from modern cancer therapies. However, author will benefit from addressing the following questions:

Major points:

Comment 1. In the abstract, authors mentioned “Biomarkers, such as cardiac troponins and natriuretic peptides, were employed to detect subclinical cardiotoxicity.”. No data relevant to biomarker are shown in this paper.

Response 1: Thanks for the observation. That's right. That part of the study has already been published. So I've deleted it from here.

Comment 2. The paper aims to investigate the association between advanced TNM stages and risk of cardiac dysfunction in patients with LVEF <50%. None of the figure is showing such data. Instead the manuscript focused on age, weight, height and body surface and quality of life.

Response: Thank you for your observation. I have corrected subtitle 3.4 and completed it with the requested information. (lines 425-436)

Comment 3. Figure 3 did not show any LVEF values, but shows age, weight, height and body surface instead.

Response 3: Thanks for the amendment. I have replaced the figures.

Comment 4. Two figures are labeled as Figure 5.

Response 4. Thanks for the amendment. I have corrected them.

Comment 5. The content related to Vitamin D in the discussion section is not relevant to this manuscript.

Response 5. Thanks for the amendment. I have deleted this part.

Minor points:

Comment 1. The demographic data shown from line 221 to line 240 are more clear to present in a graph.

Response 1: Thank you very much for suggestion. I have added a figure (line 257)

Comment 2. Pls define the exclusion group and the study group at the very beginning of the paragraph for the ease of reading.

Response 2: Thank you very much for suggestion. We moved the exclusion criteria to the beginning, next to the inclusion criteria.

Comment 3. At line 280, what is AC short for?

Response 3: Thanks for the amendment. I've added the explanations.

Comment 4.  At line 291, what is ET short for?

Response 4: Thanks for the amendment. I've added the explanations.

Comment 5.  What does FEVS mean? Does it mean LVEF? If yes, pls unify the nomenclature through the entire manuscript.

Response 5: Thanks for the amendment. I have corrected the entire manuscript.

Comment 6.  In figure 6, firstly, figure legend is unclear. A rephrasing is absolutely necessary. In addition, the label of cancer stage is confusing. What is stage 2A? Pls explain the meaning of each stage number shown in the figure. How does it corelate to TNM standards?

Response: Thanks for the amendment. I've added the explanations and I have corrected the figures.

Reviewer 2 Report

Comments and Suggestions for Authors

In this work, Murg et al. did a retrospective study analyzing the cardiac function of breast cancer patients undergoing cytostatic treatments. They found an association between advanced TNM stages and reduced left ventricular ejection fraction. This study revealed the importance of closely monitoring cardiac parameters during cancer treatments and tailored cardioprotective strategies to improve patient outcomes. However, there are still several concerns and questions that need to be addressed before the acceptance for publication:

1.     In Line 230, the authors brought the performance index to describe patients. Could the authors explain the definition and range of the performance index?

2.     The paragraph from Line 248 to Line 255 is repetitive. Could the authors combine this paragraph with previous ones?

3.     In the Differential Diagnosis section, how does each example case represent its corresponding portion of the total population? Same symptoms, disease progression, or treatments?

4.     In Line 405, how did the authors tell if the lower LVEF is associated with cancer treatment or progression?

5.     There are different expressions/labels for tumor stage (I-IV in Line 133, T1-T4 in Figure 5, 1.00-4.00 in Lines 423-426). Are they the same? If so, it will be beneficial to be consistent.

6.     Figures 5 and 6 are hard to read. What are the numbers in the graph referring to?

7.     In Lines 27 and 28, the authors mentioned that they tested global longitudinal strain and biomarking, including troponins and natriuretic peptides. Where are these data?

8.     It will be helpful to keep one abbreviation throughout the manuscript for left ventricular ejection fraction instead of two (LVEF, FEVS).

9.     Some full forms of abbreviations are missing: CORE in Line 75, CR in Line 76, and AC in Line 280.

Author Response

Reviewer 2

We sincerely thank the reviewer for their valuable feedback and constructive suggestions, which have greatly contributed to improving the quality and clarity of our manuscript. Your insights have helped us refine key aspects of our study, and we appreciate the time and effort you dedicated to providing such thoughtful and thorough comments. We have carefully addressed all your recommendations and believe the revised manuscript is significantly enhanced as a result. Thank you for your invaluable guidance and support in this process.

In this work, Murg et al. did a retrospective study analyzing the cardiac function of breast cancer patients undergoing cytostatic treatments. They found an association between advanced TNM stages and reduced left ventricular ejection fraction. This study revealed the importance of closely monitoring cardiac parameters during cancer treatments and tailored cardioprotective strategies to improve patient outcomes. However, there are still several concerns and questions that need to be addressed before the acceptance for publication:

Comment 1. In Line 230, the authors brought the performance index to describe patients. Could the authors explain the definition and range of the performance index?

Response 1: Thanks for the amendment. I've added the explanations. (lines 176-183)

Comment 2. The paragraph from Line 248 to Line 255 is repetitive. Could the authors combine this paragraph with previous ones?

Response 2: Thanks for the amendment. I have deleted this paragraph, and the informations was including in figure 2.

Comment 3. In the Differential Diagnosis section, how does each example case represent its corresponding portion of the total population? Same symptoms, disease progression, or treatments?

Response 3: Thanks for the amendment. I've added the explanations. (lines 288-291)

Comment 4. In Line 405, how did the authors tell if the lower LVEF is associated with cancer treatment or progression?

Response 4: Thanks for the amendment. I have completely reworded it.

Comment 5.     There are different expressions/labels for tumor stage (I-IV in Line 133, T1-T4 in Figure 5, 1.00-4.00 in Lines 423-426). Are they the same? If so, it will be beneficial to be consistent.

Response 5: Thanks for the observation. I've added the explanations. (lines 132-140)

Comment 6.     Figures 5 and 6 are hard to read. What are the numbers in the graph referring to?

Response 6: Thanks for suggestion. I've changed the figures and I added the explanations.

Comment 7.     In Lines 27 and 28, the authors mentioned that they tested global longitudinal strain and biomarking, including troponins and natriuretic peptides. Where are these data?

Response 7: Thanks for the observation. That's right. That part of the study has already been published. So I've deleted it from here.

Comment 8.     It will be helpful to keep one abbreviation throughout the manuscript for left ventricular ejection fraction instead of two (LVEF, FEVS).

Response 8: Thanks for the amendment. I have corrected the entire manuscript.

Comment 9.     Some full forms of abbreviations are missing: CORE in Line 75, CR in Line 76, and AC in Line 280.

 Response 9: Thanks for the amendment. I've added the explanations. (lines 176-183)

Round 2

Reviewer 1 Report

Comments and Suggestions for Authors

Thanks for addressing my comments. Authors might benefit from addressing the following minor points:

1. What does the blue and black color mean in Figure 4?

2. In figure2, the bars of body surface area in m² and performance index are invisible

Author Response

Reviewer 1

We sincerely thank the reviewer for their valuable feedback and constructive suggestions, which have greatly contributed to improving the quality and clarity of our manuscript. Your insights have helped us refine key aspects of our study, and we appreciate the time and effort you dedicated to providing such thoughtful and thorough comments. We have carefully addressed all your recommendations and believe the revised manuscript is significantly enhanced as a result. Thank you for your invaluable guidance and support in this process.

Thanks for addressing my comments. Authors might benefit from addressing the following minor points:

Comment 1. What does the blue and black color mean in Figure 4?

Response: Thank you very much for observation. I added the explanation. (lines 285-288)

Comment 2. In figure2, the bars of body surface area in m² and performance index are invisible

Response: Thank you very much for the comment. We, the authors, have converted the unit of measurement to cm² for body surface area to standardize the graphic presentation and removed the performance index, as this information is already included in the text.

Reviewer 2 Report

Comments and Suggestions for Authors

I'm satisfied with the revised manuscript. Thanks.

Author Response

Reviewer 2

We would like to express our sincere gratitude for your invaluable support and guidance throughout the review process. Your constructive feedback and thoughtful suggestions have been instrumental in enhancing the quality and clarity of our manuscript. We greatly appreciate the time and effort you dedicated to ensuring that our work meets the high standards of the journal. Thank you once again for your expertise and encouragement, which have made this process both productive and rewarding.

Sincerely,

The authors.